# p53 orchestrates DNA replication restart homeostasis by suppressing mutagenic RAD52 and POLθ pathways

Sunetra Roy[1], Karl-Heinz Tomaszowski[1], Jessica W Luzwick[1], Soyoung Park[1], Jun Li[2], Maureen Murphy[3], Katharina Schlacher[1]*

[1]Department of Cancer Biology, University of Texas MD Anderson Cancer Center, Houston, United States; [2]Department of Genomic Medicine, University of Texas MD Anderson Cancer Center, Houston, United States; [3]Molecular and Cellular Oncogenesis Program, The Wistar Institute, Philadelphia, United States

**Abstract** Classically, p53 tumor suppressor acts in transcription, apoptosis, and cell cycle arrest. Yet, replication-mediated genomic instability is integral to oncogenesis, and p53 mutations promote tumor progression and drug-resistance. By delineating human and murine separation-of-function p53 alleles, we find that p53 null and gain-of-function (GOF) mutations exhibit defects in restart of stalled or damaged DNA replication forks that drive genomic instability, which isgenetically separable from transcription activation. By assaying protein-DNA fork interactions in single cells, we unveil a p53-MLL3-enabled recruitment of MRE11 DNA replication restart nuclease. Importantly, p53 defects or depletion unexpectedly allow mutagenic RAD52 and POLθ pathways to hijack stalled forks, which we find reflected in p53 defective breast-cancer patient COSMIC mutational signatures. These data uncover p53 as a keystone regulator of replication homeostasis within a DNA restart network. Mechanistically, this has important implications for development of resistance in cancer therapy. Combined, these results define an unexpected role for p53-mediated suppression of replication genome instability.

DOI: https://doi.org/10.7554/eLife.31723.001

*For correspondence:
kschlacher@mdanderson.org

## Introduction

One of the most prominent hallmarks of cancer is genomic instability (*Hanahan and Weinberg, 2011*). As such, many DNA damage response or repair genes that restore genome stability are known tumor suppressors, including p53, the guardian of the genome (*Kim et al., 2015*). In breast cancers, p53 mutations are associated with more aggressive and triple negative breast cancers (*Turner et al., 2013*). Similar to high serous ovarian cancers, these aggressive cancers respond to chemotherapy including platinum drugs and PARP inhibitors initially, but develop resistance thereafter (*Luvero et al., 2014*; *Wahba and El-Hadaad, 2015*).

First thought to be a proto-oncogene, the initial discovery of a gain-of-function (GOF) p53 mutant allele (*Lane and Crawford, 1979*; *Levine and Oren, 2009*; *Linzer and Levine, 1979*) masked the loss of wild-type (WT) p53 function. Despite early discrepancies, only a decade later p53 was recognized as a tumor suppressor (*Baker et al., 1989*). Loss of p53 function can occur either by deletion or by mutation. Mutations may also result in a GOF, typically enhancing transcription functions. To date, the most consistent defect for both null and GOF p53 mutants in cancers is the loss of p53 transcriptional responses to regulate apoptosis and cell cycle arrest.

Genetic data of several separation-of-function p53 mutant mice suggest that there are additional p53 functions that contribute to tumor progression, which are transcription independent; Murine p53 R172P, corresponding to human R175P, retains much of its tumor suppression function despite

**eLife digest** When a cell divides to make more cells, it duplicates its DNA to pass on an identical set of genes to the new cell. Copying DNA – also known as DNA replication – is a complex process that involves several steps. First, the double helix gradually unwinds and unzips to separate the DNA strands. This creates a molecule known as the 'replication fork'. Then, copies of each strand are created and proofread for errors. Eventually, the strands are sealed back together so that the helices contain one old and one new part.

But sometimes errors sneak in during DNA replication, which can lead to mutations that may cause cancer. The higher the number of mutations, the bigger the chance is that the cancer becomes aggressive and resistant to therapy. Some of the most common mutations found in tumors happen in a protein called p53. This protein is known to stop tumors from growing by selectively killing cells with mutations. When p53 is faulty, mutant cells no longer die and can grow uncontrollably to form tumors. However, its killing abilities do not fully explain how p53 protects cells from accumulating mutations that can cause cancer, and until now, it was not known if p53 also had any other roles.

Now, Schlacher et al. discovered that p53 can protect the DNA from mutations. The experiments used normal cells and cancer cells from humans and mice, in which p53 was either blocked or modified. The experiments revealed that p53 plays an important role during DNA replication. When p53 is 'healthy', it binds to the replication fork. This ensures that replication restarts properly after it has passed faulty patches of the DNA.

The p53 protein also helps to organize the proteins involved in DNA replication. When p53 was absent or mutated, the DNA-repair protein that usually binds to the fork failed to attach properly. Instead, other proteins prone to make mutations took over the replication fork and created a pattern of mutations commonly found in tumors resistant to treatment.

A next step will be to investigate p53's role at damaged DNA replication forks and how it interacts with other proteins involved in DNA replication. To fully understand all roles that p53 plays in preventing tumor growth can help to find new ways to treat tumors with p53 defects or tumors that have become resistant to treatment.

DOI: https://doi.org/10.7554/eLife.31723.002

loss of transcriptional induction and loss of apoptosis (*Liu et al., 2004*). Similarly, p53 mutations in the transactivation domain and p53 acetylation mutations severely inhibit p53 induction of apoptosis and senescence, yet exhibit a mild and delayed tumor onset (*Li et al., 2012*; *Zhu et al., 2015*). p53 also has seemingly disparate cellular functions including during metabolism and epigenetic control, that is through its interaction with MLL3/4 histone methyltransferases (*Pfister et al., 2015*; *Zhu et al., 2015*), although the contribution of these functions to tumor suppression is not fully understood.

For cancer, a prominent p53 function is to maintain genomic stability upon DNA damage as part of a damage response. Since DNA damage traditionally is most prominently considered in the context of double-strand break (DSB) lesions, many studies focus on putative p53 functions in DSB repair. Next to error-free repair of DSBs by homologous recombination (HR) involving BRCA1/2 and RAD51, DSBs may also be repaired by non-homologous end joining (NHEJ), or through secondary and typically mutagenic pathways of single strand annealing (SSA) mediated by RAD52 and micro-homology mediated end joining (MMEJ) involving POLθ (*Black et al., 2016*; *Branzei and Foiani, 2008*; *Moynahan and Jasin, 2010*; *Wood and Doublié, 2016*). In response to DNA damage, ATM mediates p53 phosphorylation as part of a DNA stress response (*Saito et al., 2002*), which is facilitated by PTIP (*Jowsey et al., 2004*), a BRCT domain containing protein that is part of the MLL3/4 complex.

Similar to its transcription function, discrepancies ensue in the molecular function of p53 during DNA repair. While indirect studies found p53 to inhibit error-free homologous recombination (HR) and spontaneous sister-chromatid exchange (SCE), which somewhat paradoxically was proposed to promote genomic stability (*Bertrand et al., 2004*; *Gatz and Wiesmüller, 2006*), loss of p53 does not change DSB repair rates by HR when measured in specific induced break assays (*Willers et al.,*

*2001*). Thus, the mechanism by which p53 promotes genomic stability associated with tumorigenesis remains contradictory.

As previously hypothesized (*Cox et al., 2000*), recent findings formalized that 2/3 of all mutations found across cancers are caused by errors occurring during proliferation (*Tomasetti et al., 2017*), highlighting the critical importance of protective mechanisms during DNA replication. Intriguingly, early studies found p53 is activated at stalled replication forks (*Gottifredi et al., 2001*; *Kumari et al., 2004*), which are a source for genomic instability requiring distinct replication fork stability pathways (*Branzei and Foiani, 2010*). p53 interacts with BLM helicase at replication forks and represses HR in S-phase upon DNA damage, independent of its G1-S and transactivation activity (*Bertrand et al., 2004*; *Janz and Wiesmüller, 2002*; *Saintigny and Lopez, 2002*). Hinting at a direct p53 replication function, recently p53 was found to interact with DNA POLι (*Hampp et al., 2016*). Moreover, p53 deletion in U2OS cells was reported to slow unperturbed replication (*Klusmann et al., 2016*), although this was suggested to require p53 transcription function, while p53 functions in DNA damage response are not.

Here, we identify a critical role for p53 in balancing replication pathway homeostasis and show p53 suppresses replication genomic instability independent of transcription activation. We find p53 mutant alleles that separate transcription activation and replication restart functions and reveal a direct correlation between p53 replication and tumor progression functions. Importantly, we find p53 directly binds to ongoing and stalled DNA replication forks. Utilization of mutagenic RAD52/POLθ replication pathways increase for both GOF and p53 null alleles in a transcription independent manner, consistent with mutation signatures that we identify in p53 mutant breast cancers. Our results thus allow for an unexpected alternative hypothesis for acquisition of drug resistance in breast cancer cells due to p53 loss: mutant p53 boosts mutagenic RAD52/POLθ pathways, which increase deletion and point mutations that can lead to secondary resistance mutations.

## Results

### Transcription-independent p53 function for restart of stalled replication forks

p53 is an ATM phosphorylation target, is activated at stalled replication forks (*Kumari et al., 2004*) and interacts with BLM helicase. As BLM helicase is implicated in replication restart (*Davies et al., 2007*), as is ATM (*Trenz et al., 2006*), we tested the role of p53 in DNA replication reactions when stalled with dNTP depleting hydroxyurea (HU). Using single-molecule DNA fiber spreading (*Figure 1A*), we assessed the number of stalled replication forks after low-dose replication stalling (*Figure 1A*), as a test for defects in replication restart. We find a doubling of stalled forks in CRISPR/CAS9- engineered p53- null human HAP-1 cells compared to cells with wild-type (WT) p53 (*Figure 1B*; 35% stalled forks in p53 null 18% WT p53 HAP-1). This suggests a prominent role for p53 in the resumption of DNA replication after replication stress.

Increased fork stalling is classically compensated for by increased new origin firing, as seen for CHK1 defects (*Petermann and Helleday, 2010*; *Petermann et al., 2010*). Unexpectedly, we find that increased fork stalling in p53 null cells is accompanied by a decrease, rather than an increase, in new origin firing compared to both WT p53 (*Figure 1—figure supplement 1A*; 20% newly fired origins in HAP-1 cells, respectively, and 9% newly fired origins in HAP-1 p53 null cells). Taken together, the data suggests that p53 defects in HAP-1 cells exhibit distinct and unconventional replication restart defects resulting in both decreased replication restart and decreased new origin firing.

In tumors, p53 is deleted or mutated, the latter typically resulting in GOF (*Freed-Pastor and Prives, 2012*). To test whether p53 mutations alter replication restart, we investigated one of the most common GOF mutations in primary mouse embryonic fibroblasts (murine p53 R172H corresponding to human p53 R175H) (*Liu et al., 2000*). These cells show both an increase in stalled replication forks, and fewer newly fired forks compared to WT p53 MEFs (*Figure 1C*; 36% stalled forks in p53 R172H MEF and 15% in WT MEF and *Figure 1—figure supplement 1B*). These data thus uncover conserved defective outcomes for both null and GOF p53 mutations at stalled replication forks.

Murine p53 R172P corresponds to a rare human polymorphism R175P, which results in loss of transcriptional activation and apoptosis resembling p53 null (*Liu et al., 2004*). Yet, tumor

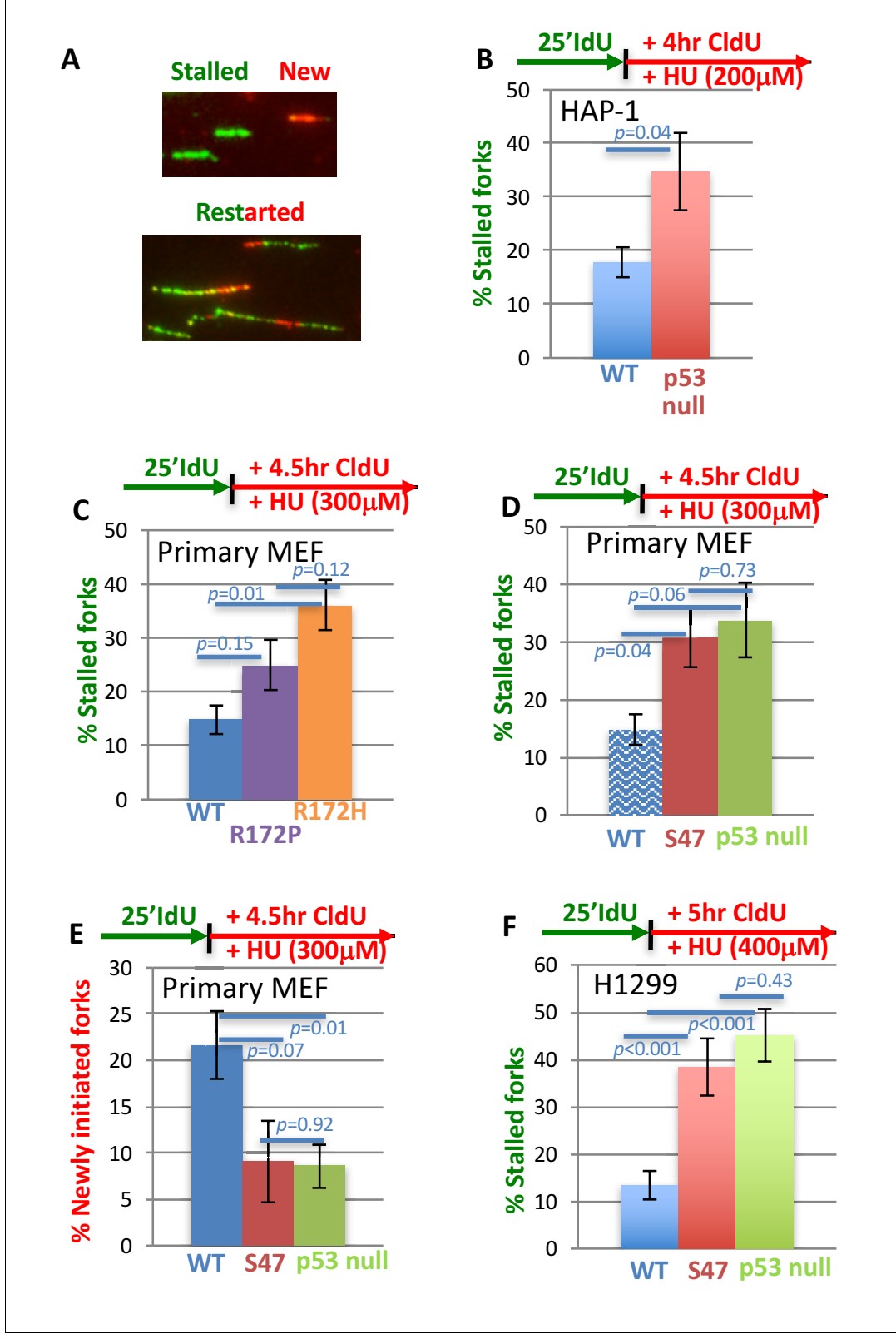

**Figure 1.** With one supplement p53 promotes efficient replication restart of stalled DNA forks. (**A**) Representative image of DNA fibers. The number of stalled forks (# stalled/(# stalled +#restarted) or newly initiated origins ((# CldU/(# stalled +# restarted +#new) is obtained in the following panels. (**B**) HAP-1 p53 null cells (**C**) primary MEF with GOF p53 R172P (apoptosis deficient, mildly tumor prone) and p53 R172H (apoptosis deficient, strongly tumor

*Figure 1 continued on next page*

*Figure 1 continued*

prone), (D, E) primary MEF with p53 S47 (apoptosis and greatly transcription proficient) and null for p53 (for better comparison WT p53 of (C) is re-plotted, shady blue), and (F) human H1299 cells with inducible p53 WT or S47. Error bars represent the SEM. Significance values are derived from student T-test analysis.

DOI: https://doi.org/10.7554/eLife.31723.003

The following figure supplement is available for figure 1:

**Figure supplement 1.** Mutant p53 exhibits unconvenional replication restart.

DOI: https://doi.org/10.7554/eLife.31723.004

---

development is markedly less severe in p53 R172P mice compared to p53 null mice, suggesting alternative mechanisms contribute to tumor progression besides transcriptional regulation of apoptosis with this mutation (*Liu et al., 2004*). We therefore tested p53 R172P MEFs and find that they only show a moderate increase in fork stalling compared to p53 R172H (*Figure 1C*). Thus, restart functions in p53 R172 mutant cells show an improved correlation with tumor suppression activity in vivo.

To further examine the possible link of failed p53-mediated replication restart and cancer, we tested p53 S47 (P47S), which is a breast-cancer pre-disposition polymorphism in African-descent populations (*Jennis et al., 2016*). It is largely transcriptionally active including for p21 (*Felley-Bosco et al., 1993*; *Jennis et al., 2016*); it remains proficient for apoptosis (*Felley-Bosco et al., 1993*; *Jennis et al., 2016*). Yet, p53 S47 mice are tumor prone, and p53 S47 contributes to breast cancer risk in African populations (*Jennis et al., 2016*; *Murphy et al., 2017*). We find that S47 slows cell growth similar to WT p53 (*Figure 1—figure supplement 1C*) consistent with intact cell-cycle check-point functions. Furthermore, we find that p53 S47 primary MEFs exhibit a loss-of-function (LOF) for replication restart, as measured by an increase in stalled forks upon HU treatment compared to WT MEF cultures (*Figure 1D*; 31% stalled forks in p53 S47 MEF versus 15% in WT MEF). Moreover, p53 S47 MEFs resemble p53 null MEFs in their inability to restart forks (*Figure 1D*; 34% stalled forks in p53 null MEF). Similar to p53 null HAP-1 and GOF R172H MEF cells, both p53 null MEF and p53 S47 MEFs show defects in new fork initiation (*Figure 1E*; 21% newly fired origins in p53 WT MEF versus 9% and 8% in p53 S47 MEF and p53 null MEF).

To test whether p53 S47 restart defects are conserved in human cells, we expressed human p53 S47 under doxycycline control in H1299 non-small cell lung carcinoma cells (*Figure 1—figure supplement 1D*) and examined stalled forks. We found an increase in stalled forks that resembles p53-null H1299 cells. Both p53 null and p53 S47 H1299 cells exhibit a substantial increase in stalled forks compared to WT p53-expressing H1299 cells (*Figure 1F*; 13% stalled forks in p53 WT, 38% in p53 S47, and 45% in null H1299 cells). Taken together, these results suggest that p53 promoted replication restart can be genetically separated from its transcription-activation function in cell cycle progression and apoptosis.

## p53 restart defects promote replication-dependent genome instability and cellular sensitivity to replication stalling agents

Restart defects cause cellular sensitivity to replication stalling agents, as seen for cells with BLM defects (*Davies et al., 2004*). We reasoned that in mutant p53 cells, this cellular phenotype so far may have been obscured by loss of apoptosis, which can override cellular sensitivity by inhibiting cell death. We therefore tested cellular sensitivity using the LOF mutant p53 S47, which remains largely apoptosis proficient. We find that p53 S47-expressing H1299 cells are sensitive to replication stalling agents HU (*Figure 2A*) and mitomycin C (MMC; *Figure 2B*). This cellular replication stress phenotype is masked when the p53 mutations additionally inactivate apoptosis and cell-cycle check-point functions, such as in p53 null H1299 cells (*Figure 2A and B*) or p53 null compared to WT mammary epithelial MCF10A cells (*Figure 2—figure supplement 1*). Collectively, the data with apoptosis-proficient p53 S47 implies that p53 functions in replication restart suppresses cellular sensitivity to replication stress.

Separation-of-function mutations p53 R172P and S47 reveal a feasible correlation between loss of p53 restart function and tumor progression. We therefore tested whether p53's replication restart function could contribute to genomic instability, which is a hallmark of cancer (*Hanahan and*

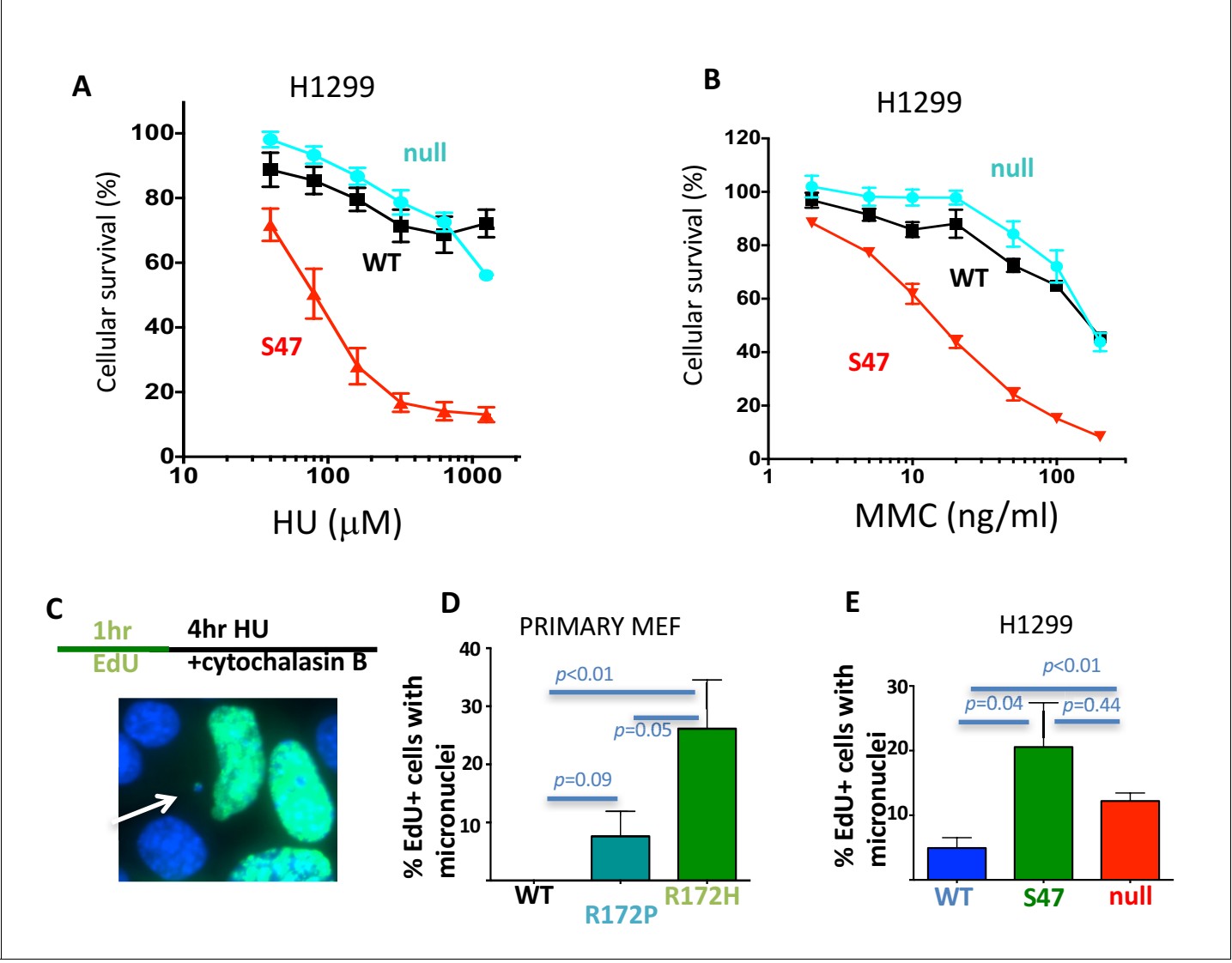

**Figure 2.** With one supplement p53 promotes replication-dependent genomic stability. (**A**) Cellular sensitivity to replication stalling with hydroxyurea (HU) in H1299 cells. (**B**) Cellular sensitivity to replication stalling with mitomycin C (MMC) in H1299 cells. (**C**) Experimental scheme and representative image of micronuclei; Scored are EdU-positive cells with micronuclei only that were in S-phase during replication stalling with HU and that are blocked during cytokinesis immediately following the offending S-phase. (**D**) Micronuclei in primary p53 R172P and R172H MEF and (**E**) human H1299 p53 WT, S47 and null cells. Error bars represent the SEM. Significance values are derived from student T-test analysis.
DOI: https://doi.org/10.7554/eLife.31723.005

The following figure supplement is available for figure 2:

**Figure supplement 1.** p53 null MCF10A cells show similar sensitivity to mitomycin C compared to WT p53 MCF10A cells.
DOI: https://doi.org/10.7554/eLife.31723.006

*Weinberg, 2011*). Amongst others, unresolved stalled replication forks can result in DNA bridges, which convert to micronuclei, a mark of BLM-defective cells (*Hoffelder et al., 2004*). We assessed genome instability by scoring micronuclei in p53 R172P, R172H and WT MEFs after arrest in cytokinesis immediately following replication stalling (*Figure 2C*). By considering EdU-positive cells only and immediate cytokinesis arrest following HU, the experimental set up ensures that only micronuclei are scored that result from induced replication stalling during the preceding S-phase. This so greatly excludes contributions of canonical G1-related p53-function. Consistent with an intermediate phenotype for replication restart, primary MEF p53 R172P exhibited less micronuclei with replication

stalling compared to p53 R172H (*Figure 2D*; average of 8% micronuclei in p53 R172P and 26% in p53 R172H) albeit considerably more than WT p53 MEFs (none detected in WT).

With replication stalling, human p53 S47-expressing cells showed a marked increase in cells containing micronuclei compared to WT p53-expressing H1299 cells (*Figure 2E* average of 20% in p53 S47% and 5% in p53 WT H1299 cells). Micronuclei instability in p53 null H1299 cells similarly is significantly higher than in WT p53 H1299 cells (*Figure 2E*; average of 12%). Taken together, the genomic instability data corresponds with restart defects rather than transcription functions found in the respective p53 mutants. The data suggest that p53 functions are required for resolving replication stress-dependent genome instability and by implication can contribute to tumor suppression.

## p53 is localized to active and stalled replication forks

p53 has a prominent DNA-binding domain and RPA interaction sites (*Romanova et al., 2004*). We reasoned that p53 could have a direct protein function during DNA replication. We therefore sought to test if p53 is present at active DNA replication forks using iPOND, which is an immunoprecipitation method of nascent EdU-labeled DNA (*Sirbu et al., 2012*). We found that p53 associates with nascent-labeled DNA in HEK293 cells (*Figure 3A*, IP fraction, E, 3.8 normalized to chromatin protein ORC2). PCNA is associated with active replication forks and therefore found reduced in iPONDs with thymidine chase after the EdU pulse (*Dungrawala et al., 2015*), (*Figure 3A*). Similarly, we find that the p53 association with nascent DNA is reduced with a thymidine chase, suggesting p53 travels with the active replication fork (*Figure 3A*). Replication stalling by HU retains p53, but not PCNA at stalled forks (*Figure 3A*). Taken together, the data suggests a direct role for p53 at active and stalled DNA replication forks.

## p53 promotes MLL3-chromatin remodeler and MRE11 restart nuclease recruitment to forks

p53 interacts with chromatin remodeling complexes and is implicated in facilitating epigenetic alterations (*Pfister et al., 2015*; *Zhu et al., 2015*). We observed an atypical restart defect in p53 mutant and null cells with less newly initiated replication forks (*Figure 1E* and *Figure 1—figure supplement 1A and B*). We hypothesized that replication restart and new replication fork firing may require local chromatin opening and epigenetic alterations mediated by p53, which could explain the unusual decrease in new fork firing with p53 mutations. To investigate local protein changes specific to replication forks, we developed the SIRF assay (in Situ Interactions at Replication Forks using PLA; *Figure 3B*). Specifically, we applied sensitive proximity ligation chemistry to detect interactions between nascent, EdU-labeled DNA and proteins within nanometer proximity. The signal is specific as elimination of EdU results in no signals (*Figure 3B*). MLL3 promotes H3K4 histone methylation to mark open chromatin (*Ruthenburg et al., 2007*). MLL3 association with replication forks in unperturbed cells is similar in p53 null and WT HAP-1 cells (*Figure 3C*; 11 MLL3-bound replication sites per cell in WT p53 and 12 sites in p53 null HAP-1 cells). Upon replication stalling with HU, we see a marked increase in MLL3-bound replication sites in WT, but not in p53 null human HAP-1 cells (*Figure 3D*; 17 MLL3-bound sites in WT and 13 in p53 null HAP-1 cells). These data suggest inefficient MLL3 recruitment to forks upon replication stalling in the absence of p53.

MLL3-mediated chromatin opening is implicated in MRE11 nuclease recruitment to stalled replication forks (*Ray Chaudhuri et al., 2016*), a repair nuclease that is needed for efficient replication restart (*Trenz et al., 2006*). We therefore examined the functional implications of reduced MLL3 recruitment to forks with p53 defects. Consistently, we find increased MRE11-bound replication sites in WT, but not in p53 null HAP-1 cells when challenged with HU (*Figure 3E* and *Figure 3—figure supplement 1*; 30 MRE11-bound replication sites/cell in WT and 18 in p53 null HAP-1 cells). Collectively, these data suggest a mechanism whereby p53 promotes local chromatin responses that aid MRE11 recruitment to stalled forks, as necessary for replication restart (*Trenz et al., 2006*).

## p53 suppresses error prone RAD52 at forks

p53 is implicated in suppressing excessive repair by homologous recombination (HR) to balance genomic stability (*Bertrand et al., 2004*; *Saintigny et al., 1999*; *Sengupta et al., 2004*). To further probe the underlying mechanism for genomic instability induced by aberrant p53 S-phase functions, we performed SIRF assays for local RAD51 recruitment to stalled forks as a surrogate marker for HR

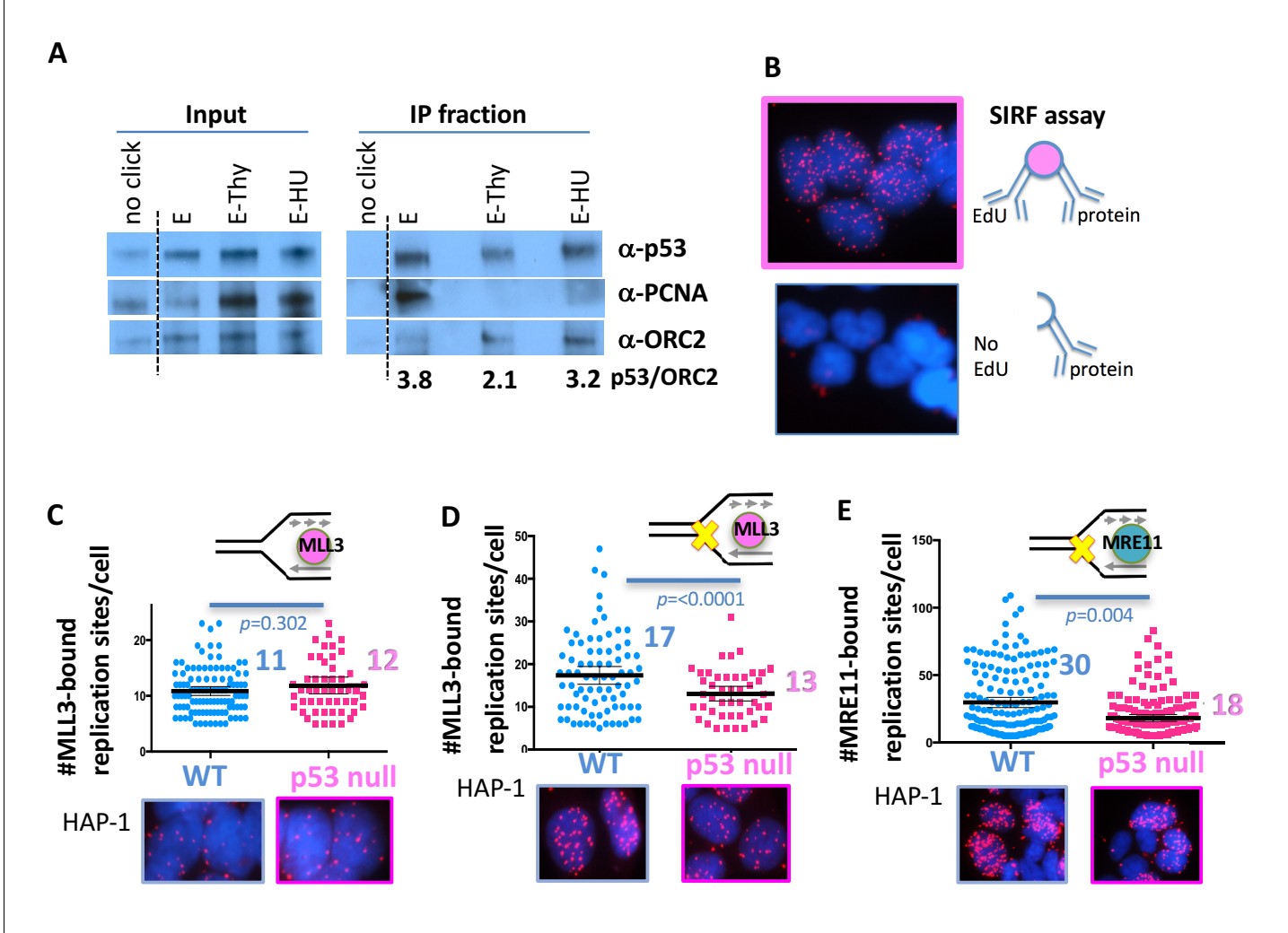

**Figure 3.** With one supplement p53 promotes recruitment of chromatin remodeler and MRE11 to stalled replication forks. (A) iPOND (immunoprecipitation of biotinylated EdU-DNA) assay of p53, PCNA and ORC2 in HEK293 cells. No click reaction omits biotin and serves as negative control. E, EdU pulse (10 μM, 10 min), E-Thy, EdU pulse followed by thymidine chase (10 μM), E-HU, EdU pulse followed by hydroxyurea (500 μM). Values are relative p53 band intensities normalized to ORC2 band intensities. (B) Schematic and representative image of SIRF (In Situ Interactions at Replication Forks) assay for interactions between protein and nascent DNA in single cells: nascent DNA is pulse-labeled with EdU a protein of interest is crosslinked to the DNA immediately following the EdU pulse. Alternatively, EdU is washed out and cells are incubated with HU (200–400 μM HU) before crosslinking. Proximity ligation assay (PLA) amplification with antibodies against EdU and the protein of interest will result in a signal only if interactions between the nascent DNA and the protein of interest are in close proximity. No signal is produced if the cell has not incorporated EdU. (C) Quantitation of SIRF assay of epigenetic remodeler MLL3 at unchallenged replication forks and (D) at HU stalled replication forks (yellow x) in HAP-1 p53 null and WT cells. (E) Quantitation of SIRF assay of MRE11 at HU stalled replication forks in HAP-1 p53 null and WT cells. Bars represent the mean and the 95% confidence interval. Significance values are derived from student T-test analysis normalized to the respective EdU-PLA intensities (*Supplementary file 1*).

DOI: https://doi.org/10.7554/eLife.31723.007

The following figure supplement is available for figure 3:

**Figure supplement 1.** Quantitation of MRE11-SIRF in unperturbed HAP-1 p53 null and WT cells.

DOI: https://doi.org/10.7554/eLife.31723.008

processes. From previous reports, we expected more RAD51 recruitment to forks in the absence of p53 (*Bertrand et al., 2004*; *Gatz and Wiesmüller, 2006*). Instead, in p53 null HAP-1 cells, we find less RAD51 at local stalled replication forks (*Figure 4A*; 24 RAD51-bound replication sites/cell in WT p53 and 19 in p53 null HAP-1 cells). In contrast, HCT116 cells expressing GOF p53 R248W show increased RAD51 fork-localization compared to WT-expressing HCT116, as do LOF mutant p53 S47-

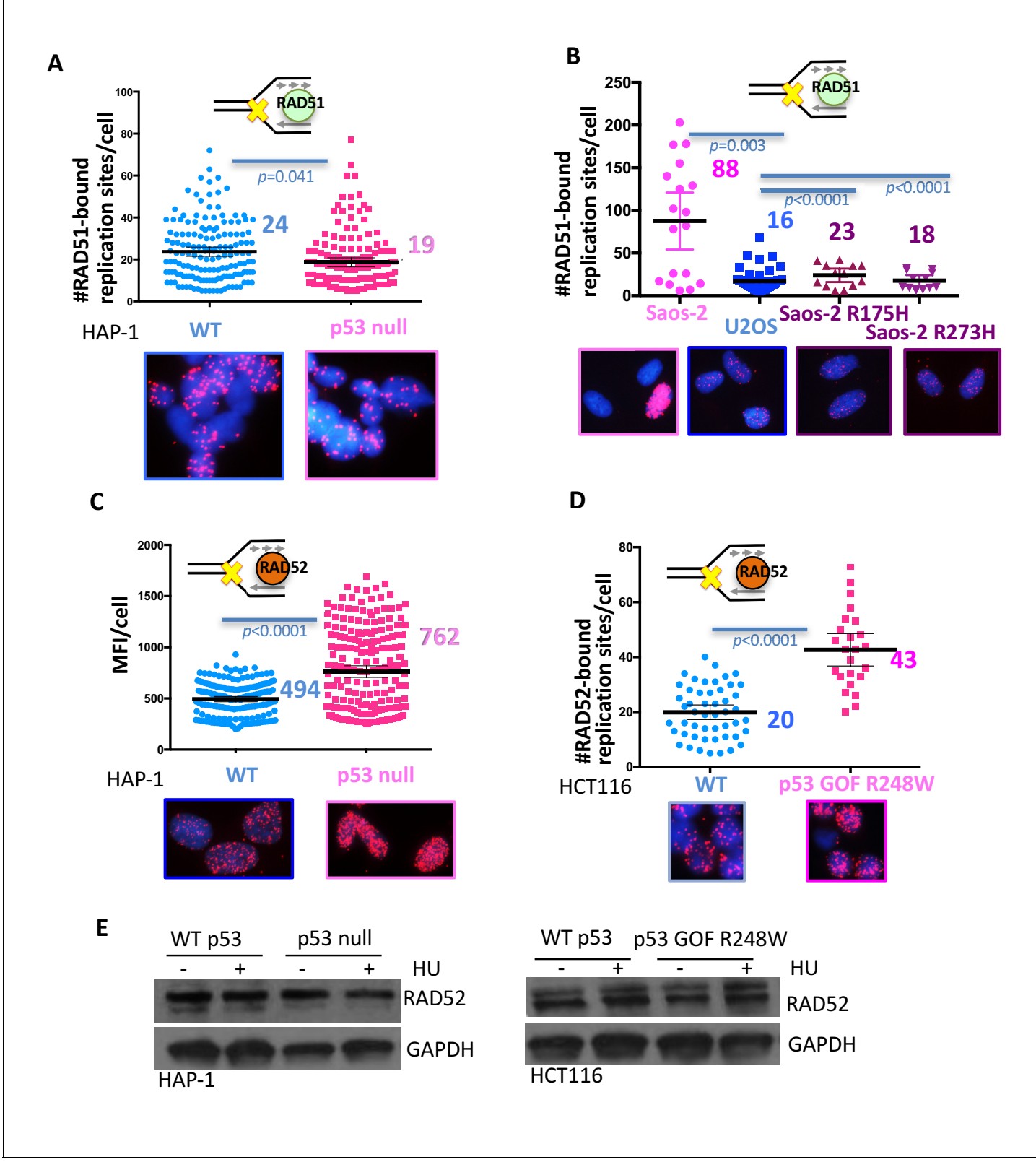

**Figure 4.** With one supplement p53 inhibits RAD52 usage at stalled replication forks. (**A**) Quantitation of SIRF assay of RAD51 at HU stalled replication forks in HAP-1 p53 null and WT cells and (**B**) p53 null, R175H and R273H complemented Saos-2 cells, and U2OS cells. (**C**) Quantitation of SIRF assay of RAD52 at HU stalled replication forks in HAP-1 p53 null and WT cells (MFI, mean fluorescence intensity) and (**D**) WT p53 or p53 R248W-expressing HCT116 cells. (**E**) Western blot of RAD52 with and without HU (200 μM) in p53 null and WT HAP-1 cells, and WT p53 or p53 R248W-expressing HCT116

*Figure 4 continued on next page*

*Figure 4 continued*

cells. Bars represent the mean and the 95% confidence interval. Significance values are derived from student T-test analysis normalized to the respective EdU-PLA intensities (*Supplementary file 1*).

DOI: https://doi.org/10.7554/eLife.31723.009

The following figure supplement is available for figure 4:

**Figure supplement 1.** RAD52-SIRF is upregulated in mutant p53 cells.

DOI: https://doi.org/10.7554/eLife.31723.010

expressing H1299 cells compared to cells with WT p53 (*Figure 4—figure supplement 1A and B*). Due to these unexpected differences, we employed p53 null Saos-2 sarcoma cells in comparison to isogenic GOF mutant p53-expressing Saos-2 cells and U2OS cells, which are p53 WT sarcoma cells (*Figure 4B*). p53 null Saos-2 show an increase in RAD51 SIRF signals, which is repressed with expression of mutant p53 GOF R175H and R273H. Together, these results do not support a correlation between RAD51 recruitment to forks and fork instability in these cells, but instead suggest alternative causation for the observed genomic instability in p53 mutant cells.

Defects in p53 do not affect HR repair of induced double-strand breaks, however, they increase spontaneous sister-chromatid exchanges (SCEs) (*Willers et al., 2001*), which are thought to occur at sites of stalled replication. Reports increasingly suggest that spontaneous SCE is independent of RAD51 and BRCA2 (*Bai and Symington, 1996*; *Clémence Claussin et al., 2017*; *Ray Chaudhuri et al., 2016*), but instead involves the single-strand annealing (SSA) protein RAD52 (*Thorpe et al., 2006*). We therefore tested whether RAD52 recruitment is altered with p53 deletions. Using SIRF analysis, we find a marked increase of RAD52 bound to stalled forks in p53-null cells (*Figure 4C*). As the signals were too abundant to be individually counted, we used the mean fluorescence intensity (MFI) as a quantitative readout (*Figure 4C*, MFI of 494 in p53 null and 762 in p53 WT HAP-1 cells).

Notably, we see stronger RAD52 recruitment at low compared to high concentration of HU (*Figure 4—figure supplement 1C*). The former condition is less favorable for DSB formation, suggesting RAD52 recruitment to stalled forks is stronger than to *bona fide* DNA breaks. Strikingly, we find an increase in RAD52 recruitment in all p53-defective cell lines tested irrespective of the nature of the p53 defect. This includes HCT116 GOF p53 R248W (*Figure 4D*), Saos-2 p53 null, Saos-2 GOF p53 R175H, Saos-2 GOF p53 R273H, and H1299 LOF p53 S47 cells (*Figure 4—figure supplement 1D and E*) compared to respective WT p53-expressing cells. These collective data unexpectedly uncover consistent replication fork pathway tipping toward mutagenic RAD52 processes in p53 defective cells. This pathway imbalance was not caused by transcriptional deregulations in p53-defective cells, as RAD52 protein levels remained unchanged with or without p53, further supporting a transcription-independent function of p53 at stalled forks (*Figure 4E*). Thus, the observed increased RAD52 recruitment to stalled forks is likely a consequence of defective replication restart.

## p53 suppresses microhomology-mediated end-joining polymerase POL*θ*

In p53-defective cells, we observed a stark RAD52 recruitment with low HU (*Figure 4—figure supplement 1C*), which can lead to reversed replication forks (*Neelsen and Lopes, 2015*) that provide free ends as substrates for DSB repair pathways. We reasoned that p53 may orchestrate reversed fork outcomes and so protect against aberrant double-strand end recognition by mutagenic DNA end pathways which may include SSA and micro-homology mediated end-joining (MMEJ). POLθ is implicated in promoting error-prone MMEJ at replication-associated DNA ends (*Roerink et al., 2014*), which includes collapsed or reversed replication forks. We therefore tested if DNA POLθ contributes to mutagenic events at imbalanced stalled forks. We find an increase of mutant p53 S47 association with POLθ in unchallenged H1299 cells, which is further enhanced with replication stalling (*Figure 5A, 24* associations per cell without and 38 with HU). Notably, WT p53-POLθ associations remain limited even with replication stalling (*Figure 5A*, average of 6 associations without and 15 with HU), suggesting pathway tipping toward mutagenic MMEJ in p53-defective cells.

To test if MRE11-dependent restart is responsible for suppression of error-prone POLθ recruitment, we inactivated the nuclease by inhibition with the specific MRE11 nuclease inhibitor PFM39 (*Shibata et al., 2014*). Inhibition of MRE11 by PFM39 greatly increased WT p53-POLθ association with replication stalling (*Figure 5A*, an increase from 15 to 50/cell average with PFM39). This

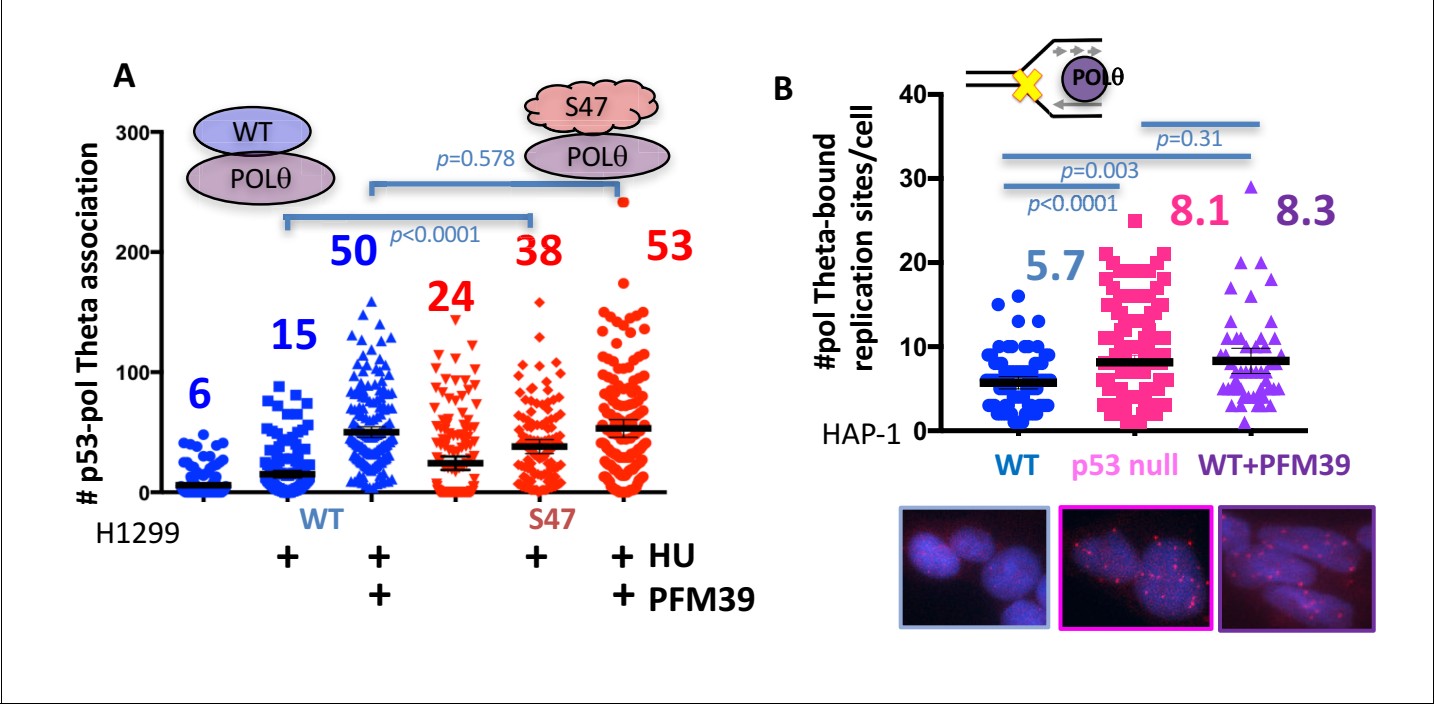

**Figure 5.** With one supplement p53 inhibits POLθ usage at stalled replication forks. (A) Quantitation of WT p53 or p53 S47 interaction with POLθ by PLA in H1299 cells with or without HU (200 µM) and MRE11 inhibitor PFM39 (100 µM). (B) Quantitation of SIRF assay of POLθ at HU stalled replication forks in HAP-1 p53 null and WT cells. Bars represent the mean and the 95% confidence interval. Significance values are derived from student T-test analysis normalized to the respective EdU-PLA intensities (*Supplementary file 1*).

DOI: https://doi.org/10.7554/eLife.31723.011

The following figure supplement is available for figure 5:

**Figure supplement 1.** PLA assay between RAD52 and POLθ in p53 R248W and WT p53 HCT116 cells.

DOI: https://doi.org/10.7554/eLife.31723.012

observation suggests that MRE11 inactivation can partially pheno-copy p53 deficiency at replication forks. In contrast, p53 S47-POLθ associations were only moderately increased with PFM39 (*Figure 5A*, increase from 38 to 53 with PFM39), where PFM39 likely blocks residual MRE11 activity in p53 S47-expressing cells. Of note, we find POLθ interactions with RAD52 increased in GOF p53 R248W HCT116 cells (*Figure 5—figure supplement 1A*), giving rise to the possibility that POLθ may collaborate with RAD52 in p53-defective cells rather than acting in separate pathways.

To further test pathway imbalance specific to local stalled forks and dependent on p53 status, we performed SIRF against POLθ in HAP-1 cells (*Figure 5B*). Consistently, we find increased POLθ recruitment to stalled forks in p53-null HAP-1 cells (*Figure 5B*). Similarly, inactivation of MRE11 nuclease in WT HAP-1 cells causes a significantly increase in recruitment of POLθ to stalled forks. Together these data uncover p53-MRE11 repression of mutagenic RAD52 and POLθ processes at replication forks.

## p53-defective breast cancers show increased mutation signatures typical for RAD52/POL*θ*

RAD52/SSA and POLθ/MMEJ pathways allow the prediction of specific mutation signatures; SSA predominantly results in larger deletion mutations, while MMEJ is signified by microhomology at repair junctions along with deletions (*Jasin and Rothstein, 2013*; *Wood and Doublié, 2016*). We therefore hypothesized that p53 replication-defective cancers may leave a telltale mutagenic pathway signature in vivo. We tested this by comparing COSMIC mutational signatures (*Stratton et al., 2009*) of p53 defective with p53 WT breast cancers reported in the TCGA database (*Figure 6*, Cosine similarity cutoff: 0.617; z-score >1.96). Seven mutational signatures are increased in p53-defective breast cancers (*Figure 6B*). However, of these seven signatures, only signatures 3 and

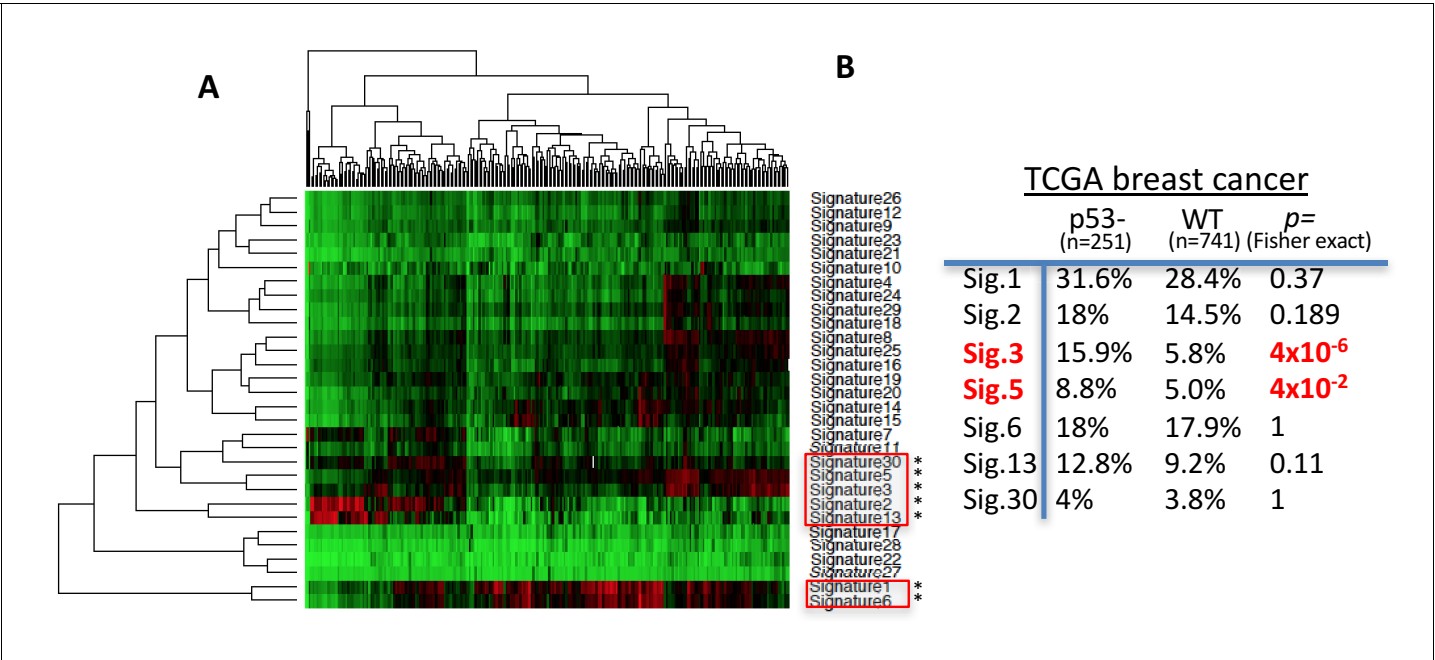

**Figure 6.** POLθ and RAD52 mutation signatures are upregulated in p53-defective breast cancers. (**A**) Hierarchical clustering of cosmic mutation signatures in p53-defective breast cancer from TCGA database. (**B**) Comparison of frequencies of cosmic signature found unregulated in p53-defective breast cancer with frequencies found in p53 proficient breast cancers. Cosmic signature in TCGA samples were analyzed by Cosine similarity; Similarity cutoff 0.617, z-score >1.96.

DOI: https://doi.org/10.7554/eLife.31723.013

The following figure supplement is available for figure 6:

**Figure supplement 1.** Cosmic mutational signatures of mutant p53 breast cancers.

DOI: https://doi.org/10.7554/eLife.31723.014

signature 5 are significantly increased in p53 defective compared to WT p53 breast cancers (*Figure 6—figure supplement 1B*); COSMIC signature 3 is defined by larger deletion mutations (>3 bp) with microhomology at break junctions, consistent with expected RAD52 and POLθ mutation spectra. COSMIC signature 5 shows T > C transition mutations at ApTpN context with thus far unknown etiology. POLθ was reported to have a stark preference for T > C transition mutations (error-rate of $42 \times 10^{-4}$, 4–40 fold higher than any other possible mutation rate) as seen within a known CA<u>T</u>CC hotspot (*Arana et al., 2008*). Thus, our combined data uncovers the possibility of POLθ mediated origin of COSMIC signature 5. Collectively, we find COSMIC signatures are in agreement with increased RAD52 and POLθ pathway usage in p53 mutant breast cancers.

## Discussion

### p53 suppresses genome instability by orchestrating replication fork homeostasis

Rather than single gene predisposition or select environmental exposure, the strongest drivers for cancer incidence are DNA replication errors. This has been long hypothesized and recently formalized by showing replication errors comprise 2/3 of all mutations in cancers (*Tomasetti et al., 2017*; *Tomasetti and Vogelstein, 2015*). This fundamental importance of replication fork maintenance is conserved in bacteria, where stress response and repair proteins primarily protect and stabilize DNA replication forks (*Cox et al., 2000*).

p53 is the 'the guardian of the genome' and the most frequently mutated tumor gene, but its functions in replication genome stability, which is the dominant source of tumor mutations, has been cryptic. The most studied p53 cellular function with regard to tumor-suppression has been its role in

transcriptional activation of apoptosis and cell cycle checkpoint. Yet, p53 functions during the DNA damage response linked to genome integrity are transcription activation independent (*Bertrand et al., 2004*; *Janz and Wiesmüller, 2002*; *Saintigny and Lopez, 2002*). Moreover, these classical p53 transactivation activities to promote apoptosis and cell cycle arrest are insufficient to fully explain p53's role in tumor suppression. This is substantiated by reported p53 separation-of-function mutations, including tumor prone yet greatly transcription activation proficient p53 mutations, such as S47. Conversely, several p53 mutant mice including p53 R175P show that inactivation of apoptosis and senescence by p53 transcription deregulation are insufficient for full inactivation of p53 tumor-suppression functions (*Brady et al., 2011*; *Li et al., 2012*; *Liu et al., 2004*). Taken together, these observations point to p53 activities in addition to its transcription activation functions that critically contribute to its tumor suppressor function. Such additional functions may include metabolism and ferroptosis, a new cell death pathway (*Li et al., 2012*; *Zhu et al., 2015*).

We here identify a new p53 function in suppressing genome instability at replication forks by promoting MLL3/MRE11-mediated replication pathway homeostasis. Importantly, this activity, which we show is independent of p53 transcription activation roles, avoids mutagenic RAD52/POLθ pathways likely acting at reversed forks (*Figure 7*). As replication mutations are thought to be the strongest cancer mutation driver and genome instability is associated with tumorigenesis, we propose that the here identified role of p53 as a replication homeostasis keeper to avoid genome instability provides a feasible novel additional p53 tumor suppression function. Moreover, the resulting understanding of p53-mediated genomic stability reconciles previous reports on apoptosis and p53 transactivation-independent roles of p53 for tumor suppression (*Phang et al., 2015*). So far, the most consistent common defect to both GOF mutant p53 and p53 gene deletion is related to its transcription function in apoptosis and cell cycle arrest. These results revealing a p53 replication-restart function reconcile how GOF and null p53 have different cellular functions and phenotypes, yet can both cause genomic instability implicated for tumor etiology and progression. Supporting this concept, MRE11 impairment, which we show phenocopies p53 defects at stalled forks, promotes progression and invasiveness of mammary hyperplasia in mouse models similar to p53 inactivation (*Gupta et al., 2013*).

Separation-of-cellular p53 function studies have spanned from apoptosis, cell cycle arrest, epigenetics and stress response to metabolism. Yet, these seemingly separate functions may act together in the context of WT p53 to guard the genome, foremost from genotoxic replication stress. As such, the replication restart function identified here conceptually connects seemingly divergent p53 functions including stress response, genome stability and epigenetics.

We propose that upon activation by replication stress, p53 orchestrates balanced error-free replication restart and suppresses genome instability, which is caused by excessive usage of mutagenic replication pathways when p53 is defective. Importantly, this model implies p53 promotes a replication homeostasis balance at forks for successful proliferation rather than a strict pathway control. If

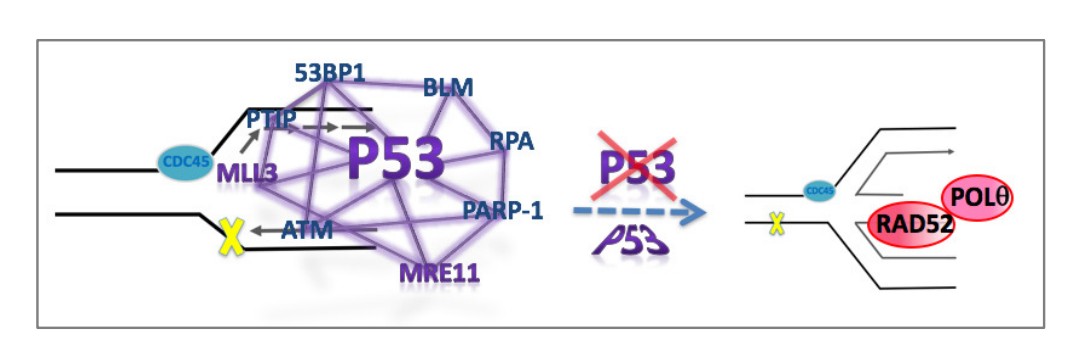

**Figure 7.** Model for p53-mediated pathway homeostasis p53 is implicated as a keystone protein that is part of a larger replication restart network. p53 mutations, defects or MRE11 defects tip the replication pathway homeostasis toward increased mutagenic RAD52/POLθpathways at unprotected stalled forks, such as reversed replication forks, and resulting in deletion and point mutations.
DOI: https://doi.org/10.7554/eLife.31723.015

replication stress exceeds a threshold for proper genome maintenance, p53 may dissociate and, as a keystone replication stress regulator, induce cell death, including but not exclusively through apoptosis, as an added safeguard to avoid cellular dysplasia.

## p53 replication fork reactions and implied biological functions

We here find the African-descent tumor variant p53 P47S (S47) to be a separation of function mutation defective in replication restart. p53 is phosphorylated by ATM at S46, which is decreased in p53 S47 (*Jennis et al., 2016*). Intriguingly, at the adjacent residues D48/D49, p53 can directly interact with single-strand binding protein RPA (*Romanova et al., 2004*), which is implicated in replication fork remodeling (*Neelsen and Lopes, 2015*). Specifically, RPA interaction mutations deregulate recombination reactions without affecting transactivation reactions (*Romanova et al., 2004*). By proximity of these residues and phenotypical commonalities, we suggest that p53 P47S (S47) may also affect RPA interactions. By extension, we propose that ATM phosphorylation of WT p53 may regulate such p53-RPA interactions for the purpose of fork remodeling, as a controlled process for restart balancing. Notably, we find p53 S47 exhibits cellular sensitivity to the DNA cross-linking reagent mitomycin C, which most prominently activates the Fanconi Anemia tumor suppressor pathway. The latest identified Fanconi Anemia tumor suppressor is the RFWD3 ubiquitin ligase that regulates p53 (*Feeney et al., 2017*; *Inano et al., 2017*). Furthermore, a Fanconi Anemia phenotypes causing patient mutation in RFWD3 leads to deregulation of RPA reactions at the replication fork (*Inano et al., 2017*). We therefore propose that p53 could feasibly be a vital player in the Fanconi Anemia pathway through its replication function, and it will be exciting to decipher this relationship.

We establish here that at forks, p53 controls MRE11, a nuclease known to promote restart after replication stalling (*Trenz et al., 2006*). Other prominent p53 collaborating proteins including BLM helicase (*Davies et al., 2007*), ATM (*Trenz et al., 2006*), and PARP-1 (*Bryant et al., 2009*) all promote replication restart. Our results thus implicate p53 as a potential keystone regulator of a greater restart network at stalled forks involving fork-reversal regulation players (*Figure 7*). PARP, BLM helicase and RPA promote and repress replication fork reversal (*Neelsen and Lopes, 2015*). While it is unclear whether fork reversal is required for normal replication restart, it is readily observed in cancer cells at low concentrations of replication stalling agents (*Zellweger et al., 2015*). In the absence of fork reversal control and stabilization by p53 regulated players, our model suggests that RAD52/POLθ pathways hijack the free DNA end to invade replication ahead or behind the replication fork as an intramolecular reaction. This could in principle lead to deletion and insertion mutations with micro-homology and increased POLθ dependent point mutations. In support of this model, we find COSMIC cancer mutation signature three signified by larger deletion mutations with micro-homologies increased in cancers with p53 defects. Additionally, we found COSMIC signature five to be increased in p53-defective breast cancers, which shows T > C transition mutations at ApTpN context with so far unknown etiology. However, our analysis shows that this signature is consistent with POLθ-mediated mutations: POLθ shows a striking preference for T > C transition mutations (error-rate of $42 \times 10^{-4}$, 4–40 fold higher than any other possible mutation rate) as seen within a known CAT̲CC hotspot (*Arana et al., 2008*). Based on our data, we therefore suggest that POLθ-mediated mutagenesis may contribute to COSMIC signature 5 etiology.

## Implications of p53 restart function in therapy resistance

In mammary tumor suppression, p53 cooperates with BRCA1/2 (*Jonkers et al., 2001*; *Ludwig et al., 1997*), which is often associated with more aggressive and resistant tumors, although the mechanism of this collaboration has been elusive. Therapy resistance in BRCA-defective cells can arise through secondary mutations. Specifically, gene internal deletion and/or point mutations within the BRCA2 gene can restore reading frames of BRCA2 mutated stop codons in CAPAN-1 pancreatic and POE ovarian cancer cells (*Sakai et al., 2009*; *Sakai et al., 2008*). Interestingly, while BRCA2 peptide expression is restored, some of these new BRCA2 peptides conferring resistance have extensive internal deletion mutations. Such deletion mutation etiology is consistent with both RAD52 and POLθ pathways providing a specific and testable mechanism for development of resistance.

Our model of RAD52/POLθ pathway increase at stalled replication forks promoting secondary mutation driving resistance is further supported by our understanding of tumor resistance biology. Triple negative breast cancers as well as serous ovarian cancers, which almost exclusively harbor p53

mutations, are often initially sensitive to therapy such as cis-platin drugs (*Luvero et al., 2014*; *Wahba and El-Hadaad, 2015*), which cause replication stalling. After multiple treatments and opportunities for RAD52/POLθ-mediated mutagenic events at stalled forks, secondary mutations can become fixed and in turn promote survival and resistance. In this scenario, mutation fixation is not necessary to promote a proliferative advantage per se. Rather it could arise from stochastic and opportunistic replication stalling events promoted by the therapeutic drug dependent on the replication program of the tissue type, consistent with both a neutral mutation evolution theory (*Sottoriva et al., 2015*) and replication errors driving tumor etiology (*Tomasetti et al., 2017*).

We here identified a new p53 role for suppressing genome instability by orchestrating balanced replication fork homeostasis. Importantly, this role is derailed in both p53 null and GOF p53 mutants, which is the only LOF ascribed to both aside from transcriptional deregulation of apoptosis and cell cycle checkpoint. Our observations and concepts reconcile prevailing paradoxes of divergent p53 functions. They, furthermore, imply specific changes in strategies for cancer patient care: our model suggests that inhibition of RAD52/POLθ pathways as adjuvant therapy concomitant with initial conventional therapy could offer an actionable strategy for ameliorating aggressive tumor evolution and secondary mutations leading to resistance in p53-defective tumors. The finding that p53 is a key-protein in error-free replication restart may explain why p53 mutations are a dominant cause of cancer genome instability.

# Materials and methods

**Key resources table**

| Reagent type (species) or resource | Designation | Source or reference | Identifiers |
|---|---|---|---|
| Cell line (human) | HAP-1 parental | Horizon Discovery | RRID:CVCL_Y019 |
| Cell line (human) | HAP-1 p53 null | Horizon Discovery | HZGHC001068C007 |
| Cell line (human) | HCT116 parental | Invitrogen | HD PAR-007, RRID:CVCL_0291 |
| Cell line (human) | HCT116 R248W GOF | Invitrogen | HD 104–003 |
| Cell line (human) | H1299 p53 null | PMID: 27034505 | |
| Cell line (human) | H1299 WT | PMID: 27034505 | |
| Cell line (human) | H1299 S47 | PMID: 27034505 | |
| Cell line (human) | U2OS | ATCC | RRID:CVCL_0042 |
| Cell line (human) | SAOS-2 | PMID: 25024203 | |
| Cell line (human) | SAOS-2 R175H | PMID: 25024203 | |
| Cell line (human) | SAOS-2 R273H | PMID: 25024203 | |
| Cell line (mouse) | MEF p53 null | PMID:14702042 | |
| Cell line (mouse) | MEF WT | PMID:14702042 | |
| Cell line (mouse) | MEF R172H | PMID:15607981 | |
| Cell line (mouse) | MEF R172P | PMID:14702042 | |
| Cell line (mouse) | MEF p53 null | PMID: 27034505 | |
| Cell line (mouse) | MEF S47 | PMID: 27034505 | |
| Cell line (human) | MCF10a | Invitrogen | HD PAR-024, RRID:CVCL_0598 |
| Cell line (human) | MCF10a p53 null | Invitrogen | HD 101–005 |
| Antibody | IdU | BD Biosciences | 347580, RRID:AB_400326 |
| Antibody | CldU | Novus Biologicals | NB500-169, RRID:AB_10002608 |
| Antibody | Mouse Biotin | Sigma | B7653, RRID:AB_258625 |
| Antibody | Rabbit Biotin | Cell signaling | 5597S, RRID:AB_10828011 |
| Antibody | MLL3 | Abcam | ab32581, RRID:AB_881525 |
| Antibody | MRE11 | Abcam | ab214, RRID:AB_302859 |

*Continued on next page*

*Continued*

| Reagent type (species) or resource | Designation | Source or reference | Identifiers |
|---|---|---|---|
| Antibody | RAD51 | Abcam | ab213, RRID:AB_302856 |
| Antibody | RAD52 | Santa Cruz | sc-365341, RRID:AB_10851346 |
| Antibody | GAPDH | Santa Cruz | sc-47724, RRID:AB_627678 |
| Antibody | Pol Theta | Abcam | ab80906, RRID:AB_1658691 |
| Antibody | PCNA | Santa Cruz | sc-56, RRID:AB_628110 |
| Antibody | ORC2 | Abcam | ab31930, RRID:AB_776911 |
| Antibody | p53 | Santa Cruz | sc-126, RRID:AB_628082 |
| Antibody | Ku70 | Cell Signaling | 4104S, RRID:AB_1904185 |
| Commercial assay or kit | Mouse plus PLA probe | Duolink Sigma | DUO92001-100RXN |
| Commercial assay or kit | Rabbit minus PLA probe | Duolink Sigma | DUO92005-100RXN |
| Commercial assay or kit | PLA detection reagent red | Duolink Sigma | DUO92008-100RXN |
| Chemical compound, drug | EdU | Invitrogen | A10044 |
| Chemical compound, drug | Hydroxyurea (HU) | Sigma | H8627 |
| Chemical compound, drug | Thymidine | Sigma | T1895 |
| Chemical compound, drug | IdU | Sigma | I7125 |
| Chemical compound, drug | CldU | Sigma | C6891 |
| Chemical compound, drug | Dynabeads MyOne streptavidin T1 | Invittrogen | 65601 |
| Chemical compound, drug | Alexa Fluor 488 Azide | Invitrogen | A10266 |
| Chemical compound, drug | Cytochalasin B | Sigma | C2743 |
| Chemical compound, drug | Formaldehyde solution | Sigma | 252549 |
| Chemical compound, drug | Paraformaldehyde 32% solution, EM grade | EMS | 15714 |
| Chemical compound, drug | Goat Serum | Sigma | G9023 |
| Chemical compound, drug | Biotin Azide | Invitrogen | B10184 |
| Chemical compound, drug | DAPI | Life Technologies | 62248 |
| Chemical compound, drug | Prolong Gold Antifade reagent | Invitrogen | P36934 |
| Chemical compound, drug | MTS | Promega | G3580 |
| Chemical compound, drug | Mitomycin C | Sigma | M0503 |
| Software, algorithm | NIS elements | Nikon | RRID:SCR_014329 |
| Software, algorithm | Duolink Analysis tool | SigmaDuolink | RRID:SCR_015574 |
| Software, algorithm | GraphPad Prism 6 | GraphPad (La Jolla, California) | RRID:SCR_002798 |
| Software, algorithm | Image J | | RRID:SCR_003070 |

## Cell lines and reagents

HAP-1 parental and HAP-1 TP53 null (Horizon Discovery) cells were grown in Iscove's modified Dulbecco's medium (Life Technologies) supplemented with 10% fetal bovine serum (Gemini Bio products) and 100 units/ml Pen-Strep (Life Technologies). H1299 small lung cell carcinoma cells expressing doxycycline inducible human WT and S47 mutant p53 constructs were previously described (Jennis et al., 2016). H1299 and HEK293 cells were grown in Dulbecco's modified Eagle medium supplemented with 10% fetal bovine serum and 100 units/ml Pen-Strep. P53 protein expression was induced by 0.5 µg/ml Doxycycline (Sigma-Aldrich). MCF10A p53 null cells were obtained from Thermo Fisher Scientific and grown in DMEM/F12 medium supplemented with 5% horse serum (Gibco), 20 ng/ml EGF (Thermo Fisher Scientific), 0.5 µg/ml hydrochortisone (Sigma-Aldrich), 100 ng/ml Cholera toxin (Sigma-Aldrich), 10 µg/ml Insulin (Sigma-Aldrich), 5 mM Hepes (Gibco), 100 units/ml Pen-Strep. MEF harboring p53 mutations R172P, R172H were previously described (Liu et al., 2004), and MEF harboring p53 mutations S47 and WT p53 and p53 null MEF were previously described (Jennis et al., 2016), and obtained from the Guillermina Lozano lab and the Maureen Murphy lab, respectively. MEFs were grown in Dulbecco's modified Eagle medium supplemented with 10% fetal bovine serum and 100 units/ml Pen-Strep and 2 mM glutamine. MEFs were generated from C57BL/6J mice with mixed sex background. HCT116 parental and CRISPR engineered mutant cells (R248W/-) were obtained from Thermo Fisher Scientific and grown in McCoy's 5a media (Lonza) with 10% fetal bovine serum and 100 units/ml Pen-Strep. Saos-2 cells complemented with GOF p53 mutants were previously described (Xiong et al., 2014), provided by Dr. Guillermina Lozano's lab and maintained in DMEM (Life Technologies) with 10% fetal bovine serum and 100 units/ml Pen-Strep. Cell lines have been authenticated by short tandem repeat (STR) profile analysis and genotyping, and have been tested for Mycoplasm (PCR). All cells were grown at 37°C and 5% $CO_2$.

## DNA fiber assay

DNA fiber spreading experiments were performed as previously described (Schlacher et al., 2011). Briefly, cells were pulsed with EdU (5–125 µM), CldU (50 µM) or IdU (50 µM), washed with PBS, and then incubated with hydroxyurea (200–400 µM) and CldU (50 µM) for 4–5 hr as indicated. The cells were harvested, resuspended in PBS and lysed on a microscope slide with lysis buffer (20 mM Tris-Cl, 50 mM SDS, 100 mM EDTA). DNA was allowed to attach for 5.5 min before spreading by gravity. Slides were fixed in methanol/acetic acid (3:1), before DNA denaturation with 2.5 N HCl and neutralization with PBS (pH 8, and subsequent pH 7.5 washes). Slides were blocked with 10% goat serum and 0.1% Triton X in PBS. IdU/CldU fibers were stained using standard immunostaining with antibodies against IdU (BrdU, Beckton Dickinson, 1:100) and CldU (BrdU, Novus Biological, 1:200) was performed before mounting slides with Prolong Gold (Invitrogen, USA). IdU/CldU Fibers were imaged using a Nikon Eclipse Ti-U inverted microscope and analyzed using ImageJ software. Between 90 and 320 fibers were scored per experiment and number of stalled forks was calculated as the number of IdU tracts (green only) divided by the number of IdU tracts plus the number of IdU-CldU tracts (green followed by red). The number of newly initiated forks was calculated as the number of CldU tracts (red only) divided by the number of IdU tracts plus the number of IdU-CldU tracts (green followed by red) plus the number of CldU tracts (red only).

## SIRF assay

Cells were pulse treated with EdU, washed two times with PBS and subsequently treated with HU (0.2 µM) for 4 hr. Cells were fixed, permeabilized with 0.25% TritonX, and a click-iT reaction was performed using biotin azide (Life Technologies) according to manufacturer's instructions. After incubation with primary antibodies, a Duolink proximity ligation assay (Sigma-Aldrich) was performed with mouse/rabbit detection red reagents according to the manufacturer's instructions. Slides were stained with DAPI and mounted with Prolong Gold before imaging using Nikon Eclipse Ti-U inverted microscope. Signals were analyzed using Duolink software, ImageJ and Nikon NIS elements, in addition to hand-counting of PLA signals. Data of repeated experiments were combined, and statistical analysis was performed using Prism6 software.

## Proximity ligation assays

H1299 cells were treated with 0.5 µg/ml doxycycline (Sigma-Aldrich) for 48 hr to induce expression of WT and mutant p53 and subsequently treated with 100 µM PFM39 (synthesized by the MD Anderson Cancer Center pharmaceutical chemistry core facility according to [*Shibata et al., 2014*]) for 30 min, followed by 0.2 mM HU for 4 hr, as indicated. Cells were fixed, permeabilized and blocked as described above and incubated with antibodies against p53 and POLθ as indicated. Finally, a Duolink PLA (Sigma-Aldrich) was performed according to manufacturer's instructions. Slides were stained with DAPI and mounted with Prolong Gold before imaging using Nikon Eclipse Ti-U inverted microscope. Signals were analyzed using Duolink software, ImageJ and hand-counted. Data of repeated experiments were combined, and statistical analysis was performed using Prism6 software.

## iPOND assay

iPOND assays were performed as described (*Dungrawala et al., 2015*). Briefly, HEK293 cells were treated with the following conditions- 10 µM EdU for 10 min, 10 µM EdU followed by 10 µM Thymidine for 1 hr and 10 µM EdU followed by 0.5 mM HU for 1 hr. Cells were subsequently fixed in 1% formaldehyde solution, quenched with glycine, permeabilized with 0.25% Triton X-100 and clicked with biotin azide as per the published protocol. Cell pellets were lysed using 1% SDS in 50 mM Tris-HCl (pH 8) and pull down was performed for 2 hr in 4°C using 50 µl/sample Dynabeads MyOne Streptavidin T1 (Invitrogen). Beads were subsequently washed once with 1 ml lysis buffer (5 min), 1 ml low-salt buffer (1% Triton X-100, 20 mM Tris [pH 8.0], 2 mM EDTA, 150 mM NaCl; 5 min), 1 ml high-salt buffer (1% Triton X-100, 20 mM Tris [pH 8.0], 2 mM EDTA, 500 mM NaCl; minutes) and finally twice with 1 ml lysis buffer (5 min). Washed beads were resuspended in 30 µl of 2X Laemmli buffer (BioRad), heated at 95°C for 25 min and proceeded for immunoblotting.

## Immunoblotting and antibodies

For western blots, cells were treated with 0.3 mM HU for 4 hr, harvested and directly lysed in Laemmli buffer (Bio-Rad), boiled for 5 min and loaded on SDS-PAGE gels.

Antibodies used for immunoblots in SIRF, PLA and iPOND are as follows: MLL3 (Abcam 1:100), MRE11 (Abcam 12D7 1:200), RAD52 (Santa Cruz F7 1:50), POL θ (Abcam 1:100), RAD51 (Abcam 14B4 1:200), mouse biotin (Sigma-Aldrich BN-34 1:100), rabbit biotin (Cell Signaling D5A7 1:200), p53 (Santa Cruz DO1, 1:1000), ORC2 (Abcam, SB46, 1:1000) and PCNA (Santa Cruz, PC10, 1:1000).

## Genomic instability assay

Cells were incubated with 50 µM EdU for 1 hr and subsequently with 0.2 mM HU and 2 µg/ml cytochalasin B (Sigma-Aldrich) for 5 hr. Cells were then collected, washed and treated with cytochalasin B for 20 hr to further capture arrested cells after division that previously were EdU labeled. Post incubation, cells were harvested and spun onto slides using a cytospin for 3 min at low acceleration setting. Cells were then fixed, permeabilized and click-iT reaction was performed with Alexa fluor 488 azide according to manufacturer's instructions. Slides stained with DAPI and mounted with Prolong Gold before imaging using Nikon Eclipse Ti-U inverted microscope. EdU-positive cells and micronuclei were scored manually and using ImageJ software. Prism was used for statistical analysis of combined repeat experiments.

## Cell survival assays

Cell viability was determined using the colorimetric MTS assay. Cells (1–2 $\times$ 10$^3$ cells) were seeded into 96-well plates for 24 hr and then exposed to varying concentrations of HU or MMC (Sigma-Aldrich) as indicated. After untreated control cells obtained ~80% confluence, the MTS assay was performed according to manufacturer's instructions (CellTiter 96 AQueous One Solution Cell Proliferation Assay, Promega). Experiments were performed in quadruplicate and repeated independently. Data was analyzed using Prism6 software and represents the mean ± standard error of the mean (SEM).

## TCGA computational analysis

The mutation annotation file (MAF) for 992 samples was downloaded from BROAD TCGA GDAC website (http://firebrowse.org/?cohort=BRCA&download_dialog=true, https://cancergenome.nih.gov). The mutation spectrum of each sample was estimated by calculating the fraction of 96 possible mutation substitutions defined in (*Alexandrov et al., 2013*) The cosine similarity score is computed for all pair-wise combinations of mutation spectrum of samples and 31 cosmic mutation signatures (http://cancer.sanger.ac.uk/cosmic/signatures). Z-score is calculated based on the distribution of all cosine similarity score ($z\_score = \frac{\cos\_score - mean(\cos\_score)}{sd(\cos\_score)}$). A z score greater than 1.96 indicates the sample could contain the corresponding cosmic signature.

## Statistical analysis

For SIRF assays, PLA signals were analyzed using Duolink Image Tool software and Nikon NIS elements software. A total of 50–300 nuclei were counted for each experimental condition. Data represents pooled experiments of two to four experiments. Signals were normalized to independent EdU-PLAs of the same condition (*Supplementary file 1*) and a T-test to determine the Z-score and p-value for significance was performed using the following equation: z = [mean (EdU-SIRF1)- mean (EdU-SIRF2)] - [mean (SIRF1)- mean (SIRF2)]/ $\sqrt{}$[Variance (EdU-SIRF1)/n + Variance (EdU-SIRF2)/n + Variance (SIRF1)/n + Variance (SIRF2)/n], whereby n is the number of measurements. The resultant p-values are indicated in the respective figures and figure legends. For DNA fiber assays, between 90 and 300 fibers were analyzed using ImageJ software. Unpaired Student t-test was performed using GraphPad Prism version six as indicated in the figures and figure legends. For genomic instability were analyzed using NIS elements software. Unpaired Student t-test was performed using GraphPad Prism version six to determine p value results as indicated in the figures and figure legends. For TCGA Computational Analaysis, Fisher Exact Test was calculated using GraphPad Prism software.

## Acknowledgements

We thank Dr. Guillermina Lozano (UT MD Anderson) for discussion and reagents (MEF p53 R172H and p53 R172P, and Saos-2 null, R175H and R273H cells). The results of *Figure 6* are based upon data generated by the TCGA Research Network: http://cancergenome.nih.gov/. We thank Dr. Ron DePinho and Dr. John Tainer (UT MD Anderson) for critical reading of the manuscript. PFM39 was synthesized by MD Anderson Cancer Center pharmaceutical chemistry core facility. This work was supported by the National Cancer Institute of the National Institutes of Health under Award K22CA175262 and by CPRIT Award R1312. KS is a Rita Allen Foundation Fellow, a CPRIT Scholar in Cancer Biology and an Andrew Sabin Family Foundation Fellow.

## Additional information

### Competing interests

Maureen Murphy: Reviewing editor, *eLife*. The other authors declare that no competing interests exist.

### Funding

| Funder | Grant reference number | Author |
|---|---|---|
| Cancer Prevention and Research Institute of Texas | R1312 | Katharina Schlacher |
| National Cancer Institute | K22CA175262 | Katharina Schlacher |

The funders had no role in study design, data collection and interpretation, or the decision to submit the work for publication.

## Author contributions
Sunetra Roy, Karl-Heinz Tomaszowski, Data curation, Formal analysis, Writing—review and editing; Jessica W Luzwick, Validation, Writing—review and editing; Soyoung Park, Data curation, Formal analysis; Jun Li, Formal analysis; Maureen Murphy, Resources, critical discussion; Katharina Schlacher, Conceptualization, Resources, Data curation, Formal analysis, Supervision, Funding acquisition, Methodology, Writing—original draft, Project administration, Writing—review and editing

## Author ORCIDs
Maureen Murphy (iD) http://orcid.org/0000-0001-7644-7296
Katharina Schlacher (iD) http://orcid.org/0000-0001-7226-6391

## Decision letter and Author response
Decision letter https://doi.org/10.7554/eLife.31723.021
Author response https://doi.org/10.7554/eLife.31723.022

# Additional files

## Supplementary files
• Supplementary file 1. EdU-PLA values for normalization of SIRFs to relative EdU incorporation.
DOI: https://doi.org/10.7554/eLife.31723.016

• Transparent reporting form
DOI: https://doi.org/10.7554/eLife.31723.017

## Major datasets
The following previously published dataset was used:

| Author(s) | Year | Dataset title | Dataset URL | Database, license, and accessibility information |
|---|---|---|---|---|
| TCGA Research Network | 2016 | Breast Cancer TCGA dataset (TCGA-BRCA) | http://firebrowse.org/?cohort=BRCA&down-load_dialog=true | Publicly available from the NCI GDC Data Portal (https://cancergenome.nih.gov) |

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
