## [Decision Letter]

Thank you for submitting your article "p53 suppresses mutagenic RAD52 and POLθ pathways by orchestrating DNA replication restart homeostasis" for consideration by *eLife*. Your article has been reviewed by three peer reviewers, and the evaluation has been overseen by a Reviewing Editor and Jessica Tyler as the Senior Editor. The following individuals involved in review of your submission have agreed to reveal their identity: Simon Powell (Reviewer #1); Thomas Helleday (Reviewer #2).

The reviewers have discussed the reviews with one another and the Reviewing Editor has drafted this decision to help you prepare a revised submission.

Summary:

All reviewers thought your paper was interesting and provocative in relation to the function of p53 in DNA replication. The paper suggests that p53 regulates replication fork stalling and restart by a mechanism that is reportedly dependent on p53's presence at or near the replication fork and the effects of p53 are transactivation independent. P53 is reported to influence the recruitment of MLL3/MRE11 and also RAD52 and Pol-theta. The mechanism of how p53 influences the recruitment of DNA repair proteins to the replication fork is presumably due to direct protein interactions of p53 at or near the fork. Overall, the paper has interest, but is not acceptable in its present form.

Essential revisions:

All reviewers now agree that a second level of evidence to put p53 at the replication fork would make the observations significantly more convincing, above and beyond the data from the SIRF assay, which is PCR-dependent. Although the SIRF assay has a link between EdU and the DNA repair protein, the location of p53 is unknown. The data raise the question about how p53 is influencing the recruitment of DNA repair proteins at the replication fork, if it is transactivation independent. Reviewers felt that i-POND could be considered, although the sensitivity to detect p53 by this method is unknown. Alternative methods to support the mechanism of how p53 influences replication fork events could be considered.

---

## [Author Response]

Summary:All reviewers thought your paper was interesting and provocative in relation to the function of p53 in DNA replication. The paper suggests that p53 regulates replication fork stalling and restart by a mechanism that is reportedly dependent on p53's presence at or near the replication fork and the effects of p53 are transactivation independent. P53 is reported to influence the recruitment of MLL3/MRE11 and also RAD52 and Pol-theta. The mechanism of how p53 influences the recruitment of DNA repair proteins to the replication fork is presumably due to direct protein interactions of p53 at or near the fork. Overall, the paper has interest, but is not acceptable in its present form.

We would like to thank the reviewers for their efforts and insightful comments on the work. We are delighted they have found the work interesting and provocative. We are pleased to submit a revised manuscript that includes additional data to address the reviewer’s recommendations. Specifically, we added iPOND data (Figure 3), made changes to the text as outline below, and added SIRF-normalization to EdU signals.

Essential revisions:All reviewers now agree that a second level of evidence to put p53 at the replication fork would make the observations significantly more convincing, above and beyond the data from the SIRF assay, which is PCR-dependent. Although the SIRF assay has a link between EdU and the DNA repair protein, the location of p53 is unknown. The data raise the question about how p53 is influencing the recruitment of DNA repair proteins at the replication fork, if it is transactivation independent. Reviewers felt that i-POND could be considered, although the sensitivity to detect p53 by this method is unknown. Alternative methods to support the mechanism of how p53 influences replication fork events could be considered.

We have now included an iPOND assay, showing that p53 localizes directly at active replication forks (Figure 3). Our data confirms and extends previous reports of p53 association with nascent EdU as seen with SILAC-iPOND (Dungrawala and Cortez, 2015). Using iPOND, we find that p53 is at ongoing replication forks, and is associated with stalled replication forks (Figure 3). We intended to convey that although the function is transcription independent, in a wild-type situation we do believe and suggest that all functions, such as transcription regulation, epigenetic regulation and replication restart, are connected for one overall biological function in maintaining replication stability. We therefore reworded our text in the Abstract, and in subsection “p53 restart defects promote replication-dependent genome instability and cellular sensitivity to replication stalling agents” to reflect functions that are “genetically separable from transcription functions” rather than describe it as a purely “transcription

independent function”.